# Method for extraction of airborne LiDAR point cloud buildings based on segmentation

**Maohua Liu[1,2]◉, Yue Shao[2]◉, Ruren Li[2]*, Yan Wang[2‡], Xiubo Sun[3‡], Jingkuan Wang[1]*, Yingchun You[2‡]**

**1** College of Land and Environment, Shenyang Agricultural University, Shenyang, China, **2** School of Transportation Engineering, Shenyang Jianzhu University, Shenyang, China, **3** Shenyang Center of Geological Survey, China Geological Survey, Shenyang, China

◉ These authors contributed equally to this work.
‡ These authors also contributed equally to this work.
\* 12216931@qq.com (RL); 13450251@qq.com (JW)

**Data Availability Statement:** The data in the article can be downloaded at the following DOI:10.6084/m9.figshare.12227945.

**Funding:** National Natural Science Foundation of China (Award Number: 51774204). The funders

## Abstract

The LiDAR technology is a means of urban 3D modeling in recent years, and the extraction of buildings is a key step in urban 3D modeling. In view of the complexity of most airborne LiDAR building point cloud extraction algorithms that need to combine multiple feature parameters, this study proposes a building point cloud extraction method based on the combination of the Point Cloud Library (PCL) region growth segmentation and the histogram. The filtered LiDAR point cloud is segmented by using the PCL region growth method, and then the local normal vector and direction cosine are calculated for each cluster after segmentation. Finally, the histogram is generated to effectively separate the building point cloud from the non-building. Two sets of airborne LiDAR data in the south and west parts of Tokushima, Japan, are used to test the feasibility of the proposed method. The results are compared with those of the commercial software TerraSolid and the K-means algorithm. Results show that the proposed extraction algorithm has lower type I and II errors and better extraction effect than that of the TerraSolid and the K-means algorithm.

## Introduction

LiDAR (Light Detection and Ranging) is an active remote sensing technology that can provide dense sets of point cloud of a scanned target. This technology has been tagged as an emerging practical technique for 3D modeling of smart cities in recent year due to its non-invasive nature, high precision, high resolution, and rapid and flexible data acquisition. The extraction of buildings from the point cloud is a prerequisite in urban 3D modeling. Current building extraction methods can be divided into two categories. One category is to directly extract the building point cloud after classifying the LiDAR data according to some features. Rottensteiner and Briese [1] combined height difference thresholds, point cloud depths, and image texture features to extract buildings. Zhang et al. [2] used filtering algorithms to separate non-ground points first and then non-ground points and corresponding near infrared. The images were overlapped, and the NDVI value was calculated for each non-ground point. Most vegetation

had no role in study design, data collection and analysis, decision to publish, or preparation of the manuscript.

**Competing interests:** The authors have declared that no competing interests exist.

points were eliminated according to the NDVI value, and the building was extracted by the European clustering algorithm based on multi-echo and area features. Cheng et al. [3] used a mathematical iterative algorithm based on reverse iteration to extract building point cloud automatically. Cao et al. [4] used the region growing algorithm of gradient threshold and area feature to extract building point cloud.

The other category is the object-oriented classification method. First, the data points are divided into multiple objects, and the divided objects are classified according to the features for extracting building point cloud. A common regional growth segmentation algorithm divides a point cloud into multiple homogeneous regions. However, some of the homogeneous regions obtained will contain other landmarks when the vegetation is adjacent to the roof of the building. Niemeyer et al. [5] provided a powerful probabilistic framework for classification based on machine learning conditional random field model and realized the point cloud extraction of buildings using random forest classifiers. Richter et al. [6] segmented the point cloud by a smooth-constrained region growing segmentation algorithm, and then used the multichannel iterative algorithm (in combination with the height difference threshold and area features) to extract building point cloud. Awrangjeb and Fraser [7] combined area, height difference, spatial position, point cloud coplanarity, and other features to extract the point cloud of buildings. Zhang et al. [8] used a region growing algorithm that combines topological, geometric, echo, and radiation properties to segment point cloud, and utilized connected component analysis and support vector machines to extract building point cloud. Li Liang [9] adopted the layer-near method to extract building point cloud. First, the region growing algorithm was used to segment the point cloud. Then, the connected component analysis was utilized to carry out the European clustering on the initial building point cloud. Finally, buildings and vegetation were distinguished by combining several features, such as area and height difference.

However, the building extraction methods based on these two ideas are complicated and need to combine multiple feature parameters. This study proposes a new building extraction method. Since the building and vegetation areas are the main components of the non-ground point elements, the normal vector direction of the building roof is basically the same, and the vegetation surface normal vector varies greatly. According to the difference of normal vector characteristics of building roof and vegetation surface, PCL based Region growth algorithm is used to segment the 3D point cloud from non-ground points. The Region growth algorithm is a curvature-based clustering algorithm, and points with curvature values within a certain threshold range are determined as the same type of features. A novel histogram method is used to distinguish buildings from non-buildings, so as to accurately extract building point cloud data. Its principle is to calculate the local normal vector and the direction cosine of the normal vector of each clustered cluster surface after clustering and generate a histogram, and extract the building point cloud based on the distribution characteristics of the histogram. Compared with currently available methods for building extraction, the proposed method is simple, efficient, and accurate. In addition, the proposed method does not need to combine any characteristic parameters and shows superiority in filtering out most of the high vegetation points and effectively extracting the building point cloud.

## Data structure and data processing methods

### Data structure

Each point of the onboard LiDAR point cloud is stored in a file in the same organizational mode because of the sensor scanning mechanism. Points in the LiDAR data are usually spatially discrete and unorganized and are unsuitable for representation on the grid due to the

geometry of the scanning device and the nature of the target interaction. Coding of each point cloud data requires all 3D coordinate values (X, Y, and Z), and the random point cloud becomes difficult to use when performing such operations as searching or performing interpolation operations. Thus, this study uses a K-D tree data structure. K-D tree is a way to organize point sets by using geometric index. It is a multi-dimensional generalization of spatial binary tree. When K-D tree method is used to divide point cloud data, the data on both sides of the node are usually divided into two parts. The number needs to be consistent, and the division ends when the number of nodes is less than the set threshold. The use of K-D tree method to organize point cloud data can greatly improve the efficiency of data search [10].

## CSF algorithm

Point cloud filtering separates the set of terrain surface points from the airborne LiDAR point cloud data. Its essence is the initial classification of point cloud data. The original airborne LiDAR point cloud is divided into ground and non-ground points, which is the key step of subsequent point cloud data processing [11]. The CSF [12,13] method is different from the traditional point cloud data filtering algorithm and it is a 3D computer graphic algorithm based on cloth simulation. The principle of the algorithm is that a piece of virtual cloth sinks on the surface of the terrain due to gravity. The cloth is assumed to be sufficiently soft enough to adhere to the surface, and the shape of the cloth is the digital surface model. The terrain is flipped over. If the cloth has rigidness and is defined by rigidness, then the final shape of the cloth is the digital elevation model. The cloth is essentially a mass spring model. By analyzing the interaction between the cloth node and the corresponding point cloud data, the shape of the cloth is finally determined to realize data filtering and obtain ground and non-ground points. The specific process of the filtering method is as follows:

(1) Invert the original point cloud data;

(2) Place the cloth above the original laser point cloud to be processed and set cloth simulation parameters;

(3) Perform grid division on the original point cloud data, and then search for the nearest laser point matching each cloth grid point to determine whether it is a movable point. If it cannot be moved, calculate the elevation of this point as a collision point.

(4) If it is a movable point, calculate the position of the point under external force and calculate the height difference between the point and the collision point. If the height difference is less than or equal to 0, the elevation of the point is set as the collision point elevation value, and it is set as the immovable point;

(5) Calculate the displacement that needs to be adjusted for each movable cloth grid point under the internal force;

(6) Repeat (4) and (5) until the number of iterations is not less than the maximum number of iterations, and the iteration ends;

(7) Determine whether the height difference between the cloth grid point and the original laser point cloud matching it is less than the threshold. If it is less than the height difference threshold, it is a ground point, otherwise it is a non-ground point.

## PCL-based region growing segmentation algorithm

PCL is an open source point cloud library that involves the acquisition, filtering, segmentation, registration, and feature extraction of point clouds. This study is based on the PCL to implement the segmentation algorithm for region growing. The segmentation algorithm is applicable to the point cloud data with uniform features. By setting constraints and combining the fusion requirements of the segmentation data, different objects are segmented from the scene

by using the features of the objects in the scene. In the PCL, the region growing segmentation class pcl::Region Growing is adopted to implement the region growing segmentation algorithm. The goal of the algorithm is to merge adjacent points satisfying the smoothing constraint, and each cluster point set belongs to the same smooth surface. The algorithm is a segmentation based on the angle difference of the normal angle. By comparing the angle between the seed point and its neighbor point, the point cloud smaller than the smoothing threshold is used as the part of the same smooth surface [14]. The points are sorted from small to large according to the curvature value of the point, and the initial seed point is selected as the point with the smallest curvature to start growing. Growing from the smoothest region reduces the total number of segments, which in turn increases segmentation efficiency. The region where the initial seed point is located is the smoothest region. The point cloud in the adjacent range of the seed point is searched, and the curvature between the seed point and each adjacent point is calculated. If the curvature is less than the set threshold, then the neighborhood point is regarded as the homogeneous region of the seed point, and the current seed point is deleted. The aforementioned steps are performed cyclically until the points in the point set are all processed, and the growth of the region ends.

### Histogram-based building detection method

The building cluster must be separated from the tall vegetation to extract the buildings in the scene. A simple method is to extract the building point cloud through LiDAR data visualization software manually, but this method is inefficient and unsuitable for large areas. Existing building extraction algorithms need to combine multiple characteristic parameters and the calculation process is complex. For example, it is necessary to combine characteristic parameters such as the height difference threshold, the point cloud depth, image texture, echo, and area. In this study, a simple and efficient histogram building detection method was proposed. The local normal vector of point cloud computing in each cluster was obtained after region growth segmentation and the direction cosine in the X, Y, and Z directions were calculated. The number of point cloud with cosine value was counted and the histogram was generated. According to the difference of histogram distribution between vegetation and building clusters, the architectural complex and non-building cluster are separated.

### Commercial software TerraSolid detection

The filtering function of TerraSolid software is based on the irregular triangulation encryption algorithm proposed by Axelsson [15]. The original point cloud is filtered to separate non-ground points, including buildings, vegetation, artificial facilities, and vehicles. The ground point elevation is used as a reference to classify non-ground points by comparing the elevation information of the point cloud and setting a reasonable elevation threshold [16]. Point clouds from 0 m to 0.5 m above ground are generally classified as low vegetation, point clouds from 0.5 m to 2 m above ground are classified as medium vegetation, and point clouds above 2 m are classified as tall vegetation and buildings.

### Filter experiment

This study selects two datasets in the south and west parts of Tokushima, Japan. Dataset 1 has a high terrain in the west and a low terrain in the east. The main ground features include buildings, tall trees, artificial facilities, low and medium vegetation, bridges, and rivers. The buildings are scattered, and the surrounding vegetation is sparse. A total of 2663447 laser foot points are present in the survey area. The point cloud density is 31 points/m$^2$. The minimum and maximum elevations are +100.80 m and +308.09 m, respectively. The original point cloud

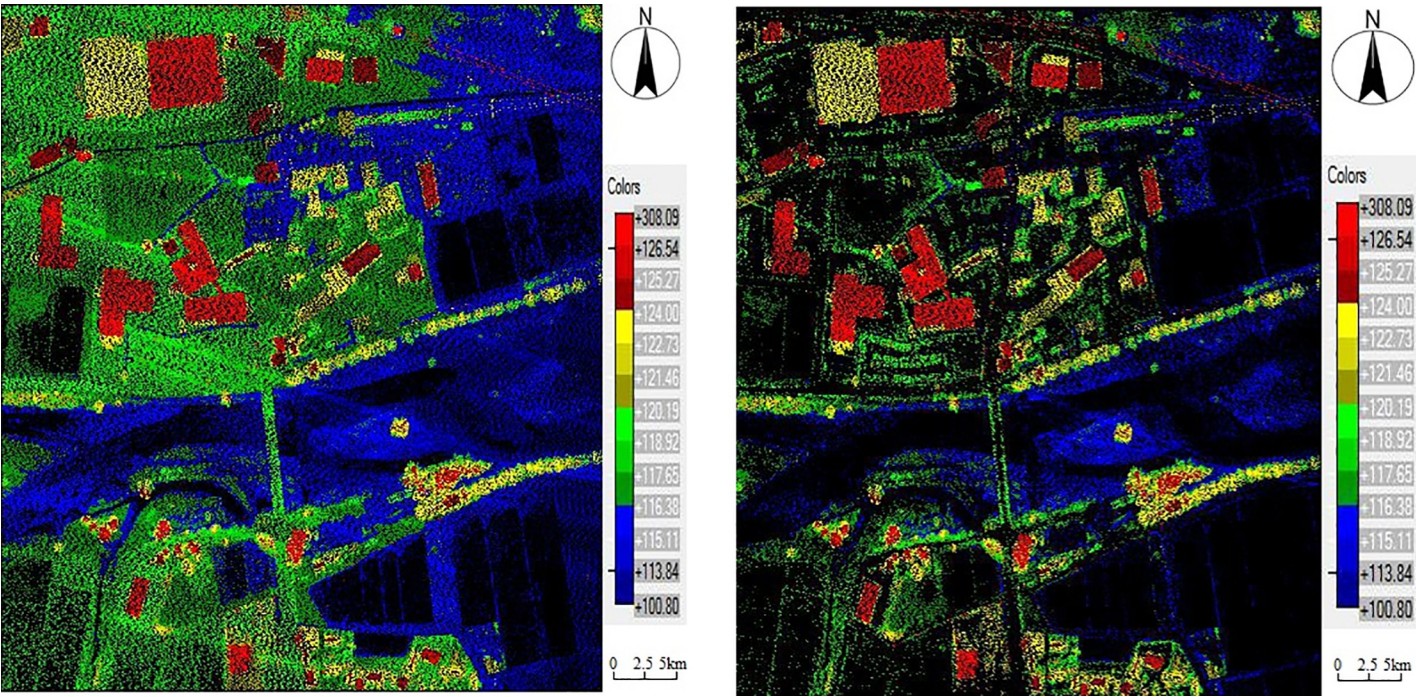

**Fig 1. Comparison of dataset 1 before and after filtering.** 1(A) Raw data before filtering. 1(B) Non-ground points after filtering.

of dataset 1 colored by elevation is shown in Fig 1(A). In dataset 2, the terrain in the survey area is low on the west and high on the east. A few types of ground objects, mainly buildings and vegetation, are concentrated and surrounded by green vegetation. A total of 2064906 laser foot points are present in the survey area. The point cloud density is 20 points/ $m^2$. The minimum and maximum elevations are +144.22 m and +355.19 m, respectively. Each color in the color legend of Figs 1 and 2 corresponds to the corresponding elevation value. The original point cloud of dataset 2 colored by elevation is shown in Fig 2(A). Filtering experiments on the point cloud data in datasets 1 and 2 are performed by the CSF algorithm to obtain non-ground points. The non-ground points of datasets 1 and 2 are shown in Figs 1(B) and 2(B), respectively. The number of points in datasets 1 and 2 before and after filtering is counted. The statistical results are shown in Table 1.

## Results and analysis

From the above-mentioned filtering experiments, the non-ground points of datasets 1 and 2 are obtained. The filtered dataset is organized by a K-D tree, and the region growing algorithm is implemented on the basis of an open-source PCL. The segmentation algorithm divides these points into different clusters. The region growing algorithm needs to set the minimum and maximum numbers of clustering points for datasets 1 and 2. According to the prior knowledge of the data, the minimum number of clustering points in dataset 1 is set to 20 and the maximum number of clustering points is set to 7200. The minimum number of clustering points in dataset 2 is set to 40 and the maximum number of clustering points is set to 5000. The normal vector of each point cloud from the obtained clusters is calculated, and the normal vector and the cosine of X, Y, and Z directions are calculated. The histograms is generated, and the building and non-building point groups are separated according to the change characteristics of the

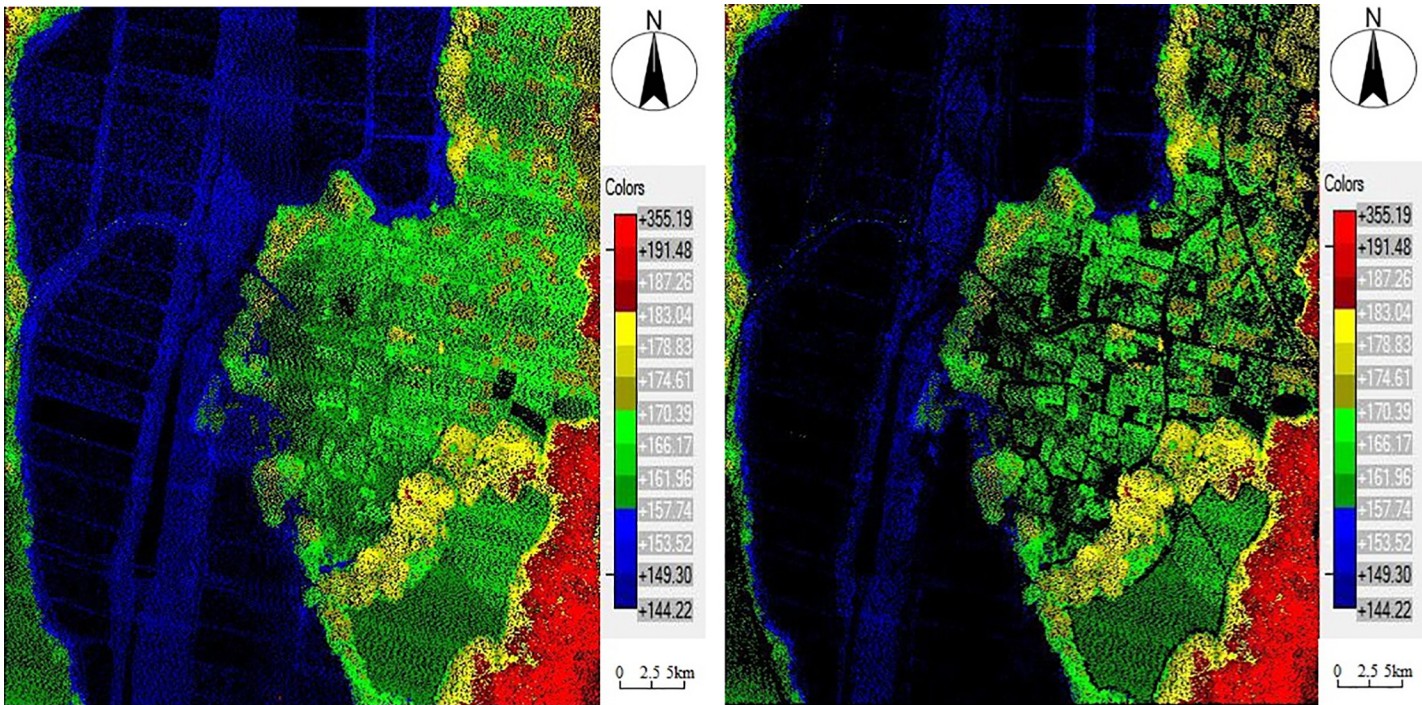

**Fig 2. Comparison of dataset 2 before and after filtering.** 2(A) Raw data before filtering. 2(B) Non-ground points after filtering.

histogram. This study enumerates the normal vector cosine of some sampling points. The histograms generated by datasets 1 and 2 are shown in Figs 4 and 5, respectively.

As shown in the histograms in Figs 3 and 4, the histogram of normal lines on the tree surface is scattered with many peaks either for datasets 1 or 2. Moreover, the normal cosine value of the maximum frequency in the X and Y directions tends to be 0, and the normal cosine value of the maximum frequency in the Z direction tends to be 1. However, the histogram of normal building surface shows few peaks and has a concentrated distribution, which can distinguish buildings from non-buildings.

Fig 5 shows the building point cloud extracted from datasets 1 and 2 by using the proposed method. A total of 65 and 105 buildings are extracted from datasets 1 and 2, respectively.

The marked area in the figure refers to the point cloud where non-buildings are mistakenly divided into buildings or incomplete point clouds are extracted. Notably, two large vehicles are mistakenly divided into buildings in dataset 1, which may be caused by that the top of the car is similar to the surface of the building and has a relatively close direction cosine value. Many data are mistakenly extracted in dataset 2, among which three are incomplete and caused by tall vegetation covering the top of buildings. In the five other places, tall vegetation adjacent to

**Table 1. Statistics of cloud points before and after filtering.**

| Parameter | Dataset 1 | Dataset 2 |
|---|---|---|
| Total points | 2663447 | 2064906 |
| Point cloud density (P/m$^2$) | 31 | 20 |
| Ground point | 1476770 | 896180 |
| Non-ground point | 1186519 | 1708640 |
| Noise point | 158 | 86 |

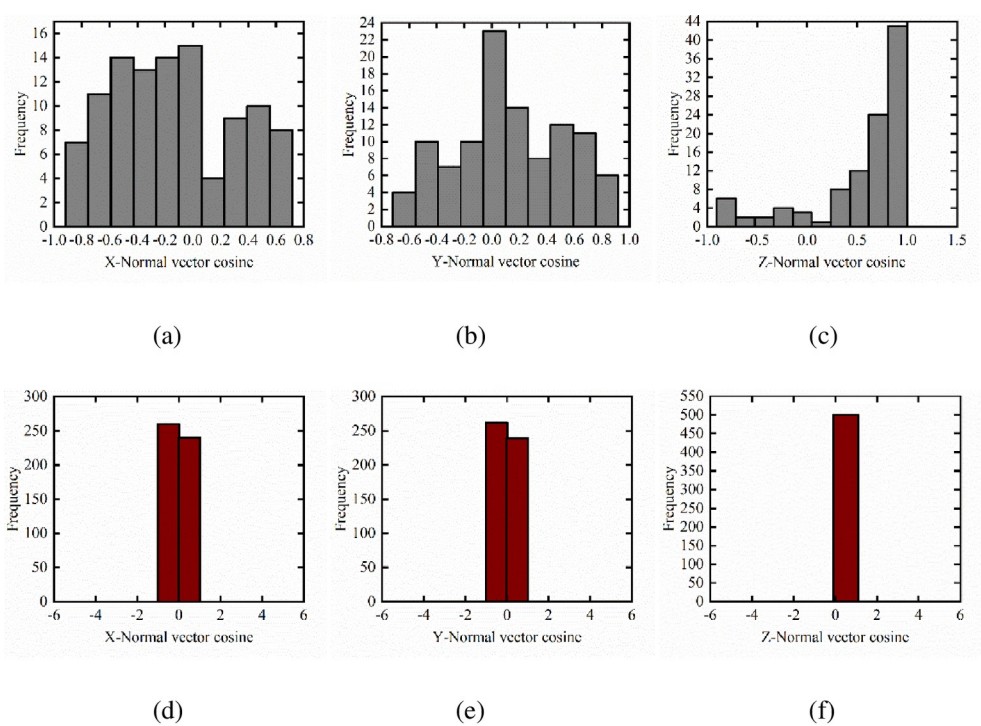

**Fig 3. Non-building and building normal histograms for dataset 1.** (a) X-tree normal histogram (b) Y-tree normal histogram (c) Z-tree normal histogram (d) X-building normal histogram (e) Y-building normal histogram (f) Z-building normal histogram.

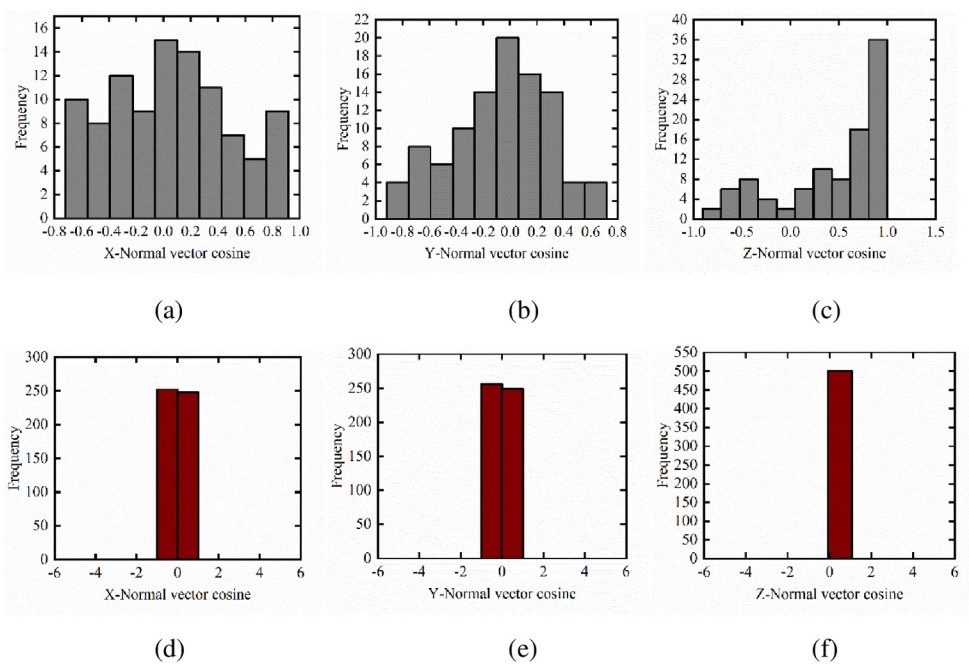

**Fig 4. Non-building and building normal histograms for dataset 2.** (a) X-tree normal histogram (b) Y-tree normal histogram (c) Z-tree normal histogram (d) X-building normal histogram (e) Y-building normal histogram (f) Z-building normal histogram.

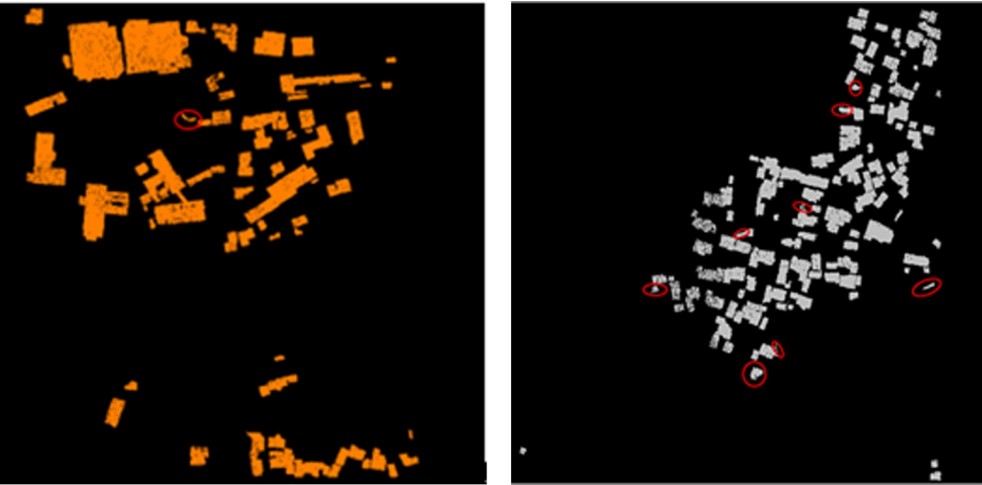

**Fig 5. Building extraction results.** 5(A) Dataset 1 building extraction results. 5(B) Dataset 2 building extraction results.

buildings is mistakenly divided into buildings. The reason is that the surface of these vegetation areas is relatively smooth. Most of the external contours are also regular, which is similar to the direction cosine of the buildings.

The building extraction based on TerraSolid software needs to set the minimum size of the building and the elevation threshold from the ground point. The minimum size of buildings in dataset 1 is 20, and the minimum size of buildings in dataset 2 is 40. Therefore, the parameters in dataset 1 are set to 20 and 2.2 m, and those in dataset 2 are 40 and 2.2 m. The building extraction results for datasets 1 and 2 are shown in Fig 6. A total of 74 and 107 buildings are extracted from datasets 1 and 2, respectively. The marked areas in the figure are all points where the vegetation points are calculated as buildings. Notably, the TerraSolid software is not ideal for extracting buildings from datasets 1 and 2.

In order to objectively evaluate the method of this paper, this paper also uses the K-means clustering algorithm based on Python to extract buildings. The K-means clustering algorithm

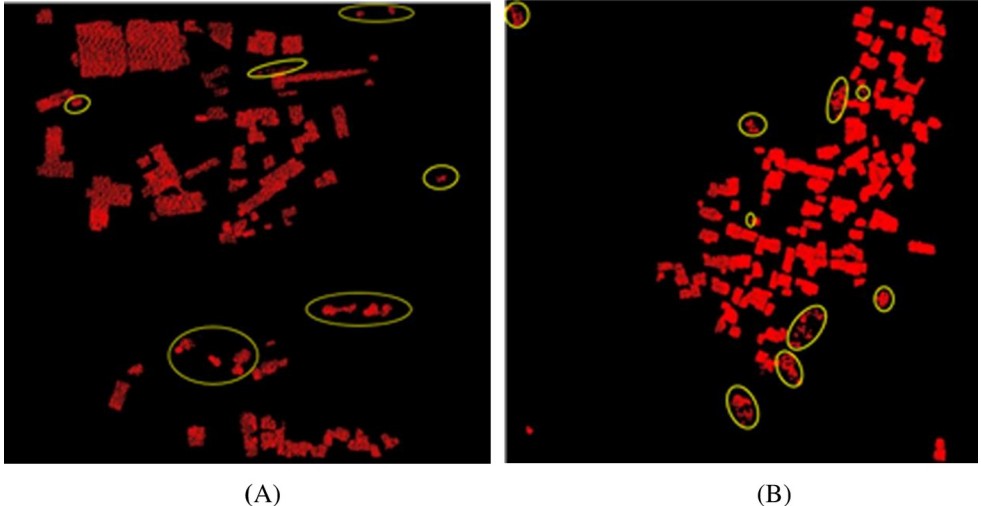

(A)  (B)

**Fig 6. TerraSolid building extraction results.** 6(A) Dataset 1 extraction results. 6(B) Dataset 2 extraction results.

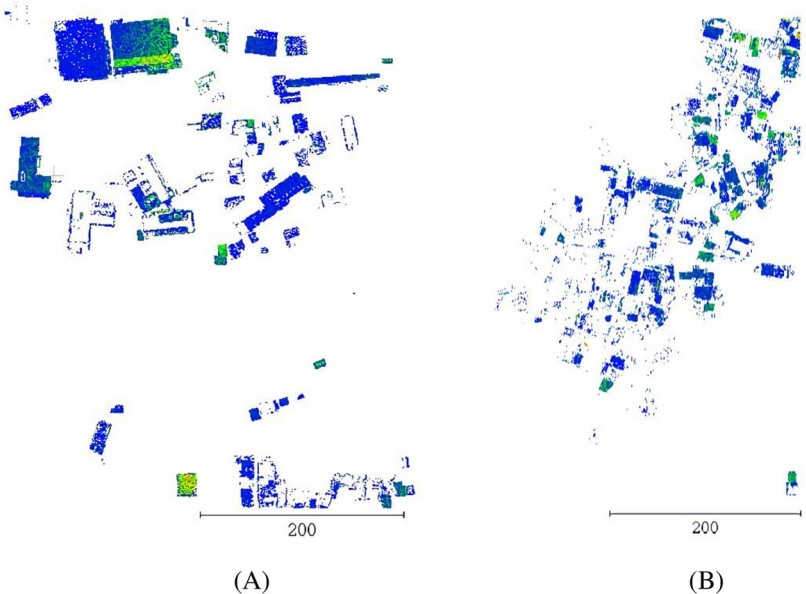

(A) (B)

**Fig 7. K-means building extraction results.** 7(A) Dataset 1 extraction results. 7(B) Dataset 2 extraction results.

is an iterative clustering analysis algorithm. The step is to randomly select K objects as the initial cluster center, then calculate the distance between each object and each seed cluster center, and assign each object to the cluster center closest to it. The cluster centers and the objects assigned to them represent a cluster. For each sample assigned, the clustering center of the cluster is recalculated based on the existing objects in the cluster. This process is repeated until a certain termination condition is met. In this paper, a total of 4 cluster classifications are set up, and the classification is based on Z value and echo intensity. The extraction results of buildings in datasets 1 and 2 are shown in Fig 7. Among them, 56 buildings are extracted in datasets 1, and 114 buildings are extracted in datasets 2. It can be seen from the figure that the buildings in the datasets 1 are incompletely extracted and only the outer contours are extracted for most of the buildings. There are many cases where non-buildings are misclassified into buildings in dataset 2. The finishing effect is not satisfactory.

In order to verify the extraction accuracy of this method, quantitative analysis was performed on the method in this paper, the method based on TerraSolid software extraction, and the K-Means method. The reference for accuracy evaluation is the results of manually extracted buildings. Three evaluation variables are defined as type I errors, type II errors and type Total errors. The type I error indicates the probability of misclassifying a building point cloud into a non-building point cloud. The type II error indicates the probability of misclassifying non-building point clouds into building point clouds. Define N as the actual number of point clouds in the experimental area, $N_1$ as the number of actual building point clouds in the experimental area, $N_2$ as the number of building point clouds extracted through experiments, and n as the number of point clouds for correctly judging buildings. The calculation formulas of the type I error, the type II error and the type Total error are shown in formula (1), formula (2) and formula (3), and the specific calculation results are shown in Table 2.

$$\text{Type I} = \frac{N_1 - n}{N_1} \tag{1}$$

**Table 2. Error analysis of point cloud extraction method for buildings.**

| Error type | Type I | | Type II | | Type Total | |
|---|---|---|---|---|---|---|
| Method | Dataset 1 | Dataset 2 | Dataset 1 | Dataset 2 | Dataset 1 | Dataset 2 |
| Proposed method | 5.68% | 8.56% | 0.36% | 1.18% | 3.66% | 5.31% |
| TerraSolid | 7.64% | 8.89% | 1.27% | 1.21% | 5.46% | 5.89% |
| K-means | 10.42% | 15.39% | 13.67 | 16.79 | 10.36% | 14.71% |

$$\text{Type II} = \frac{N_2 - n}{N - N_1} \tag{2}$$

$$\text{Type Total} = \frac{N_1 + N_2 - 2n}{N} \tag{3}$$

It can be seen that the type I errors, type II errors, and type Total errors extracted by datasets 1 and 2 using TerraSolid software and K-means algorithm are higher than that of the method in this paper. The K-means algorithm has the largest type I error, type II error and type Total error in both datasets 1 and 2. Generally, the method of this paper has the highest accuracy in extracting buildings.

## Conclusions

This study proposes a building point cloud extraction method based on PCL region growth algorithm and histogram. First, the segmentation algorithm of Region growth is used to segment non-ground points into different clusters. Then, the normal vector of point cloud in each cluster and the normal vector cosine of the X, Y, and Z directions are calculated and the histogram is generated. Experiments show that the algorithm is simple in principle and does not need to combine any characteristic parameters. Additionally, the proposed algorithm can filter out most of the high vegetation points and effectively extract the building point cloud to meet the precision requirements of building point cloud extraction. The method proposed in this paper is compared with TerraSolid, a commercial software, and the open-source algorithm K-means. For the two datasets, the mean type I error extracted by the proposed method is 7.12%, the mean type II error is 0.77%, and the total error is 4.99%. The mean values for the three types of errors for using the TerraSolid are is 8.27%, 1.24%, and 5.68%, whereas those values for the K-means algorithm is 12.91%, 15.23%, and 12.54%. In summary, the building extraction results based on the algorithm proposed in this paper have less error and higher accuracy.

## Acknowledgments

Thanks to the data provided by the Shenyang Geological Survey of the China Geological Survey and SHEN YANG KIMOTO INDUSTRIES CO.,LTD.

## Author Contributions

**Conceptualization:** Maohua Liu, Ruren Li.

**Data curation:** Yue Shao.

**Formal analysis:** Yue Shao.

**Methodology:** Jingkuan Wang.

**Supervision:** Yan Wang, Xiubo Sun.

**Validation:** Yingchun You.

**Writing – original draft:** Jingkuan Wang.

**Writing – review & editing:** Jingkuan Wang.

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
