## [Decision Letter · Decision Letter 0]

10 Jan 2020

PONE-D-19-29944

Method for extraction of airborne LiDAR point cloud buildings based on segmentation

PLOS ONE

Dear Mr. LI,

Thank you for submitting your manuscript to PLOS ONE. After careful consideration, we feel that it has merit but does not fully meet PLOS ONE’s publication criteria as it currently stands. Therefore, we invite you to submit a revised version of the manuscript that addresses the points raised during the review process.

We would appreciate receiving your revised manuscript by Feb 24 2020 11:59PM. To enhance the reproducibility of your results, we recommend that if applicable you deposit your laboratory protocols in protocols.io, where a protocol can be assigned its own identifier (DOI) such that it can be cited independently in the future. For instructions see: http://journals.plos.org/plosone/s/submission-guidelines#loc-laboratory-protocols

We look forward to receiving your revised manuscript.

Kind regards,

Tayyab Ikram Shah, Ph.D.

Academic Editor

PLOS ONE

Journal Requirements:

2. Please amend your Data availability statement to provide direct links/URLs or contact details to where other researchers can obtain the datasets. Please also provide in your methods section enough detail about the datasets so that another researcher can replicate the datasets.

3.****In your Data Availability statement, you have not specified where the minimal data set underlying the results described in your manuscript can be found. PLOS defines a study's minimal data set as the underlying data used to reach the conclusions drawn in the manuscript and any additional data required to replicate the reported study findings in their entirety. All PLOS journals require that the minimal data set be made fully available. For more information about our data policy, please see http://journals.plos.org/plosone/s/data-availability.

5. We note that Figures 2, 3, 6 and 7 in your submission contain satellite images which may be copyrighted.

a. You may seek permission from the original copyright holder of Figures 2, 3, 6 and 7 to publish the content specifically under the CC BY 4.0 license. 

6. Thank you for stating the following financial disclosure:

 'Author Contributions: LIU Maohua and LI Ruren conceived the research concept and designed it. SHAO Yue implemented the experiment and produced and analyzed the results. WANG Yan and SUN Xiubo supervised. SHAO Yue drafted the manuscript. WANG Jingkuan and YOU Yingchun reviewed the manuscript'

Please provide an amended Funding Statement that declares *all* the funding or sources of support received during this specific study (whether external or internal to your organization) as detailed online in our guide for authors at http://journals.plos.org/plosone/s/submit-now Please state what role the funders took in the study.  If any authors received a salary from any of your funders, please state which authors and which funder. If the funders had no role, please state: "The funders had no role in study design, data collection and analysis, decision to publish, or preparation of the manuscript."

Reviewers' comments:

Reviewer's Responses to Questions

**Comments to the Author**

1. Is the manuscript technically sound, and do the data support the conclusions?

Reviewer #1: Yes

Reviewer #2: Yes

2. Has the statistical analysis been performed appropriately and rigorously? 

Reviewer #1: No

Reviewer #2: No

3. Have the authors made all data underlying the findings in their manuscript fully available?

Reviewer #1: No

Reviewer #2: Yes

4. Is the manuscript presented in an intelligible fashion and written in standard English?

Reviewer #1: Yes

Reviewer #2: Yes

5. Review Comments to the Author

Reviewer #1: Good research. However, the manuscript does not sufficiently explain the scientific significance of the research presented in the manuscript. Some of my observations are given below:

• The research presented in the manuscripts does not seems to be adding any value as the results are not much improved from the commercially available TerraSolid.

• Line 197-198: No reason explained for setting different minimum – maximum clustering points for the two data sets.

• Line 238-240: No reason explained for setting different parameters for the two data sets.

• Figure 1 does not add any significant value to the manuscript and may not be required.

• Figure 2 and 3 does not have essential cartographic (Scale and north arrow) elements shown on them. No significant improvement in building is seen because of the filtering process and thus does not seems to be adding any value in the manuscript.

Reviewer #2: The authors:

This is an important work underlying methods development using LeDAR. The method established here is compared with a commercially available algorithm. I would recommend a comparison to an open source method in addition to the commercial software. I have my comments as a word document attached please.

Regards,

The reviewer

6. PLOS authors have the option to publish the peer review history of their article (what does this mean?). If published, this will include your full peer review and any attached files.

Reviewer #1: No

Reviewer #2: No

---

## [Author Response · Author response to Decision Letter 0]

3 Apr 2020

Dear Editors and Reviewers,

Thanks very much for the time and effort you spent in reviewing our manuscript entitled “Method for extraction of airborne LiDAR point cloud buildings based on segmentation” (ID: PONE-D-19-29944R1). We would like to thank the Editors-in-Chief for giving us a chance to resubmit the paper, and also thank the Associate Editor and the two reviewers for giving us constructive and insightful suggestions. These comments would be very helpful in improving the quality of our paper and provide important guidance to our future research. 

We studied the comments carefully and revised the manuscript in accordance with the comments of the editor and reviewers exactly. Revised portions were marked in red in the revised manuscript. The following is a point-to-point response to the comments of the Associate Editor and the two reviewers.

………………………………………………………………………………………

Academic Editor: 

Comment 1: Please ensure that your manuscript meets PLOS ONE's style requirements, including those for file naming. The PLOS ONE style templates can be found at http://www.plosone.org/attachments/PLOSOne_formatting_sample_main_body.pdf and http://www.plosone.org/attachments/PLOSOne_formatting_sample_title_authors_affiliations.pdf

Response: Thank the Academic Editor very much for his/her constructive comments. We have ensured that the manuscript meets the journal style requirements.

Comment 2: Please amend your Data availability statement to provide direct links/URLs or contact details to where other researchers can obtain the datasets. Please also provide in your methods section enough detail about the datasets so that another researcher can replicate the datasets.

Response: Thank the Academic Editor very much for his/her constructive comments. We have modified the data availability statement in the system and provided the URL in the cover letter.

Comment 3: In your Data Availability statement, you have not specified where the minimal data set underlying the results described in your manuscript can be found. PLOS defines a study's minimal data set as the underlying data used to reach the conclusions drawn in the manuscript and any additional data required to replicate the reported study findings in their entirety. All PLOS journals require that the minimal data set be made fully available. For more information about our data policy, please see http://journals.plos.org/plosone/s/data-availability.

Response: Thank the Academic Editor very much for his/her constructive comments. We have resubmitted the minimal dataset for the study and provided the URL in the cover letter.

Comment 4: PLOS requires an ORCID iD for the corresponding author in Editorial Manager on papers submitted after December 6th, 2016. Please ensure that you have an ORCID iD and that it is validated in Editorial Manager. To do this, go to ‘Update my Information’ (in the upper left-hand corner of the main menu), and click on the Fetch/Validate link next to the ORCID field. This will take you to the ORCID site and allow you to create a new iD or authenticate a pre-existing iD in Editorial Manager. Please see the following video for instructions on linking an ORCID iD to your Editorial Manager account: https://www.youtube.com/watch?v=_xcclfuvtxQ

Response: Thank the Academic Editor very much for her constructive comments. We have ensured that the corresponding author have an ORCID iD (0000-0002-8732-3413)and verified it in the Editorial Manager.

Comment 5: We note that Figures 2, 3, 6 and 7 in your submission contain satellite images which may be copyrighted.

Response: Thank the Academic Editor very much for his/her constructive comments. Figures 2, 3, 6 and 7 are not satellite images. They are 3D point clouds and we took a screenshot from a certain perspective for easy reading. Therefore, these figures do not involve copyright issues. Figures 2 and 3 show the results before and after data set 1 and data set 2 filtering. Figure 6 is the result of building extraction by the method in this paper. Figure 7 shows the results of building extraction based on TerraSolid.

Comment 6: Thank you for stating the following financial disclosure:

 'Author Contributions: LIU Maohua and LI Ruren conceived the research concept and designed it. SHAO Yue implemented the experiment and produced and analyzed the results. WANG Yan and SUN Xiubo supervised. SHAO Yue drafted the manuscript. WANG Jingkuan and YOU Yingchun reviewed the manuscript'

Response: Thank the Academic Editor very much for his/her constructive comments. We have provided a revised Statement of Funding and have added a revised statement to the cover letter (Highlighted part).

………………………………………………………………………………………

Reviewer #1: 

Comment 1: Line 197-198: No reason explained for setting different minimum – maximum clustering points for the two data sets.

Response: Thanks a lot for the very insightful comment. Because the data of data set 1 and data set 2 are different, the minimum and maximum number of clustering points should be set according to the specific conditions of the data. We have explained in the revised manuscript why the minimum and maximum clustering point settings in data set 1 and data set 2 are different, line 237-240.

Comment 2: Line 238-240: No reason explained for setting different parameters for the two data sets.

Response: Thanks very much for the valuable comment. TerraSolid software needs to know the minimum size of the buildings in the dataset to extract the buildings. According to prior knowledge, the minimum size of buildings in dataset 1 is 20 square meters, and the minimum size of buildings in dataset 2 is 40 square meters. We have added explanations in the revised draft, line 279-282.

Comment 3: Figure 1 does not add any significant value to the manuscript and may not be required.

Response: Thanks very much for the valuable suggestion. According to the comment of Reviewer 1, we deleted figure 1.

Comment 4: Figure 2 and 3 does not have essential cartographic (Scale and north arrow) elements shown on them. No significant improvement in building is seen because of the filtering process and thus does not seems to be adding any value in the manuscript. 

Response: Thanks a lot for the very constructive and insightful comment. Based on the comment of Reviewer 1, we have added a north arrow and a scale on Figures 2 and 3, line 224-228(Since Figure 1 has been deleted, the original Figures 2 and 3 are the current Figures 1 and 2). I want to explain here, Figures 1 and 2 here are the results before and after filtering, and the purpose is to separate ground points and non-ground points rather than extract the buildings. Non-ground points include buildings, trees, and other targets. As shown in Figures 1 and 2, most ground points are removed after filtering. Figure 5 is the result of extracting buildings (line 276) and we can clearly see the buildings in Figure 5. 

………………………………………………………………………………………

Reviewer #2: 

This is an important work underlying methods development using LiDAR. The method established here is compared with a commercially available algorithm. I would recommend a comparison to an open source method in addition to the commercial software. I have my comments as a word document attached please.

Response: Thank Reviewer 2 a lot for his/her very constructive and insightful comments and approval of our study. Reviewer 2 pointed out the major and minor problems of our study. We revised the manuscript in accordance with his/her comments exactly. According to the comment of Reviewer 2, we compared the proposed method with the K-means algorithm in the revised manuscript in addition to the commercial software (TerraSolid).

Comment 1: Page 2.Line 21-22, Suggestion: Delete “The airborne ………..to …… ground object. It is also

Response: Thanks very much for the helpful comment. We have deleted “The airborne LiDAR technology is a new type of mapping for quickly acquiring 3D data of ground and ground objects. It is also an important”.

Comment 2: Page 2.Line 22, Change suggested: The LiDAR technology is a means of …………..

Response: Thanks a lot for the very insightful and helpful comment. We have changed "It is also an important "to "The LiDAR technology is a" based on the comment of Reviewer 2, line 30. 

Comment 3: Page 2.Line 35, Cloud has lower type I …. Instead of, less type I.

Response: Thanks very much for the valuable comment. In the revised manuscript, we have changed “less type I” to “lower type I”, line 42.

Comment 4: Page 2.Line 35, Terra Solid, a commercial software.

Response: Thanks very much for the valuable comment. Since we have introduced that Terra Solid is a commercial software in the abstract, we changed the sentence to “Results show that the proposed extraction algorithm has lower type I and II errors and better extraction effect than that of the TerraSolid and the K-means algorithm.” based on the comment of Reviewer 2.

Comment 5: Page 2-4. Introduction needs strengthening, particularly on why this method is needed, and its comparative advantages to currently available open source methods. 

Response: Thanks very much for the helpful comment. According to the comment of Reviewer 2, we added some contents in the first and last paragraphs in the Introduction, line 92-94, 96-103(Highlighted content).The major superiority method over currently available methods are that the proposed method is simple, efficient, and accurate. In addition, the proposed method does not need to combine any characteristic parameters and shows superiority in filtering out most of the high vegetation points. 

Comment 6: Suggestion: Include a section to describe the study area, so as to have a clear view of the test sites and location of these two.

Response: Thanks very much for the valuable comment. The location and data characteristics of the study area have been described in the original text, lines 204-210.

Comment 7: Points in the LiDAR …… instead of The points in the LiDAR

Response: Thanks a lot for the very insightful comment. We have changed “The points in the LiDAR” to “Points in the LiDAR”, line 107.

Comment 8: ….. and unorganized. Suggestion: delete “….. and area unsuitable for …..to ….. interaction. 

Response: Thanks a lot for the very insightful comment. Considering the comment of Reviewer 2, we have deleted “and are unsuitable for representation on the grid due to the geometry of the scanning device and the nature of the target interaction”.

Comment 9: This section (Data structure) needs improvement for clear understanding of the data used. It has missing links.

Response: Thanks a lot for the very insightful comment. This section writes the organization of the point cloud rather than a detailed description of the experimental data. The specific description of the data used is in the article 204-210.

Comment 10: Suggestion: A comparison with other methods, specifically open source would be nice. 

Response: Thanks a lot for the very insightful comment. In order to better verify the usability of the method in this paper, we have added the open-source algorithm K-means algorithm to the paper. The algorithm principle and experimental results are shown in lines 290-331(Highlighted content).

Comment 11: Page 5-6.Line 97, this section needs explanation and clarity, particularly of the principle of the method.

Response: Thanks very much for the helpful comment. We added the flow of the cloth simulation filtering algorithm, lines 133-153.

Comment 12: Page 6.Line 115-117, Common ……… to region growing segmentation, etc. is an incomplete sentence. Re-write please.

Response: Thanks very much for the helpful comment. We deleted “Common segmentation algorithms such as RANSAC (Random Sampling Consistency), Euclidean distance segmentation, region growing segmentation, etc.” And the first sentence is modified as “PCL is an open source point cloud library that involves the acquisition, filtering, segmentation, registration, and feature extraction of point clouds”, line 155-156.

Comment 13: Page 7. Line 144, Suggestion: provide examples and nature of complexity.

Response: Thanks a lot for the very helpful comment. We have provided examples and nature of complexity, line 183-185.

Comment 14: Page 7.Line 147,….in each cluster was obtained after…….

Response: Thanks a lot for the very helpful comment. We have changed “….in each cluster obtained after…….” to “….in each cluster was obtained after…….”, line 187.

Comment 15: Page 9.Line 166, Confusing sentence, revise please. The data …… to …. low right.

Response: Thanks a lot for the very helpful comment. We have changed “The data set 1 shows the terrain is left high and low right.” to “Data set 1 has a high terrain in the west and a low terrain in the east”, line 205.

Comment 16: Page 9.Line 170, 31 points/m2; write and else where as points/m2.

Response: Thanks a lot for the very helpful comment. We have checked the full text and changed all points / m2 to points /m2.

Comment 17: Page 9.Line 174, Change: lush to green… surrounded by green vegetation……

Response: Thanks very much for the valuable comment. Considering the comment of Reviewer 2, we have changed “are concentrated and surrounded by lush vegetation” to “are concentrated and surrounded by green vegetation”, line 213.

Comment 18: Page 9.Line 163. As the heading of the section is “Experiment and result analysis”, I would suggest to remove this heading here.

Response: Thanks a lot for the very helpful comment. We have deleted this heading.

Comment 19: Page 11.Line 217. … to be 0…. To be 1…..

Response: Thanks a lot for the very helpful comment. We have changed this sentence in the text to “Moreover, the normal cosine value of the maximum frequency in the X and Y directions tends to be 0, and the normal cosine value of the maximum frequency in the Z direction tends to be 1”,line 258-260.

Comment 20: Page 12.Line 226. Vehicles instead of cars be more appropriate here.

Response: Thanks a lot for the very helpful comment. We have changed “cars” to “vehicles”, line 267.

Comment 21: Page 12.Line 239-240. Why the thresholds are different between set 1 and set 2.

Response: Thanks a lot for the very insightful comment. TerraSolid software needs to know the minimum size of the buildings in the dataset to extract the buildings. According to prior knowledge, the minimum size of buildings in dataset 1 is 20 square meters, and the minimum size of buildings in dataset 2 is 40 square meters. We have added explanations in the revised draft, line 279-282. 

Comment 22: Page 12.Line 242. Points are calculated as buildings. Instead of mistakenly divided into buildings…..

Response: Thanks a lot for the very insightful comment. We have changed “points are mistakenly divided into buildings” to “points are calculated as buildings”, line 285.

Comment 23: Page 13.Line 249-258. Reference data used? Or how the errors were calculated and what comparison were made?

Response: Thanks a lot for the very insightful comment. The reference data used in this paper for accuracy assessment are manually extracted building results, line 311-312.

Comment 24: Page 13.Line 261. Table 2: Also include total error in addition to Type I and Type II errors.

Response: Thanks a lot for the very insightful comment. We have added a calculation method for the Total error, as shown in equation (3). And re-calculate the method of this paper, based on TerraSolid software and K-means algorithm, Type I errors, Type II errors and Type Total errors, as shown in Table 2.

Comment 25: Page 14.Line 270. Region growth…..

Response: Thanks very much for the valuable suggestion. We have changed “region growing” to “Region growth”, line 334-335.

Comment 26: Page 14.Line 278. Is compared with TerraSolid, a commercial software.

Response: Thanks very much for the valuable suggestion. We have changed “The proposed method is compared with the commercial software TerraSolid” to “The proposed method is compared with TerraSolid, a commercial software”, line 342.

Comment 27: Figure 1.Needs clarity. It doesn’t reflect the phenomena.

Response: Thanks very much for the valuable suggestion. Taking into account the comments of Review 1 and Review 2, we decided to delete Figure 1.

Comment 28: Figure 2. Doesn’t explain …

Response: Thanks very much for the valuable suggestion. Data set 1 in Figure 2(Original Figure 2 is now Figure 1) is the original point cloud data before filtering and the non-ground point data after filtering. We have already written in the article, lines 218-220. Colors in the figure represents the size of the elevation value corresponding to each color, and we have added an explanation of Colors in the text, line 215-217.

Comment 29: Figure 4 and 5. These needs explanation or reflection of building and non-building segments. The color or pattern may be changed.

Response: Thanks very much for the insightful suggestion. We have changed the colors of the normal vector histograms of the buildings in the original Figures 4-5, see Figures 3-4 after the changes.

………………………………………………………………………………………

In addition, we proofread the manuscript to minimize the typographical, grammatical, and bibliographical errors. Therefore, some sentences and words were rewritten for a native expression, but the meanings did not change. These revisions were not included in the point-to-point response to the comments of the reviewers. However, all revised portions were marked in red in the revised manuscript.

We appreciate for the warm work of the Editors & Reviewers earnestly, and hope that the corrections will meet with approval. Once again, thank you very much for your thoughtful comments and suggestions. Should you have any questions, please contact us without hesitation. Thanks and best regards!

Yours Sincerely,

Ruren LI

February 20, 2020

---

## [Decision Letter · Decision Letter 1]

22 Apr 2020

Method for extraction of airborne LiDAR point cloud buildings based on segmentation

PONE-D-19-29944R1

Dear Dr. LI,

We are pleased to inform you that your manuscript has been judged scientifically suitable for publication and will be formally accepted for publication once it complies with all outstanding technical requirements.

With kind regards,

Tayyab Ikram Shah, Ph.D.

Academic Editor

PLOS ONE

Additional Editor Comments (optional):

Reviewers' comments:

Reviewer's Responses to Questions

**Comments to the Author**

1. If the authors have adequately addressed your comments raised in a previous round of review and you feel that this manuscript is now acceptable for publication, you may indicate that here to bypass the “Comments to the Author” section, enter your conflict of interest statement in the “Confidential to Editor” section, and submit your "Accept" recommendation.

Reviewer #2: All comments have been addressed

2. Is the manuscript technically sound, and do the data support the conclusions?

Reviewer #2: Yes

3. Has the statistical analysis been performed appropriately and rigorously? 

Reviewer #2: Yes

4. Have the authors made all data underlying the findings in their manuscript fully available?

Reviewer #2: Yes

5. Is the manuscript presented in an intelligible fashion and written in standard English?

Reviewer #2: Yes

6. Review Comments to the Author

Reviewer #2: Thank you for agreeing to the suggestions and incorporating most of the comments. Though I have no more comments, yet a few minor changes will further improve this manuscript. I have these as:

Introduction: non-invasive instead of noninvasive

Line 204, 218-219: datasets instead of data sets

Line 205, 211, 217: dataset instead of data set

211, 216, 218, 220 and elsewhere: figure instead of Fig; Check figure numbers 1(A) and then 1-2. The later be either 1(B) or the earlier be 1-1; be consistent in numberingfigure instead of Fig; check figure numbers 1(A) and then

Line 225, 226: (A) Raw data before filtering

(B) Non-ground points after filtering

Line 228: (B) Non-ground points after filtering

Line 302: dataset 1 not data sets 1. Check for this typo elsewhere in the manuscript.

Wish the authors best of luck

The Reviewer

7. PLOS authors have the option to publish the peer review history of their article (what does this mean?). If published, this will include your full peer review and any attached files.

Reviewer #2: No

---

## [Editor Report · Acceptance letter]

4 May 2020

PONE-D-19-29944R1 

Method for extraction of airborne LiDAR point cloud buildings based on segmentation 

Dear Dr. Li:

I am pleased to inform you that your manuscript has been deemed suitable for publication in PLOS ONE. Congratulations! Your manuscript is now with our production department. 

With kind regards,

on behalf of

Dr. Tayyab Ikram Shah 

Academic Editor

PLOS ONE